# Information theoretic measures of neural and behavioural coupling predict representational drift

Kristine Heiney [1,2,3,4]*, Mónika Józsa[1], Michael E. Rule[1,5], Henning Sprekeler[4], Stefano Nichele[3,6], Timothy O'Leary[1]*

1 Department of Engineering, University of Cambridge, Cambridge, United Kingdom, 2 Department of Computer Science, Norwegian University of Science and Technology, Trondheim, Norway, 3 Department of Computer Science, Oslo Metropolitan University, Oslo, Norway, 4 Modelling of Cognitive Processes, Technical University of Berlin, Berlin, Germany, 5 School of Engineering Mathematics and Technology, University of Bristol, Bristol, United Kingdom, 6 Department of Computer Science and Communication, Østfold University College, Halden, Norway

* kh732@cam.ac.uk (KH); tso24@cam.ac.uk (TL)

## Abstract

In many parts of the brain, population tuning to stimuli and behaviour gradually changes over the course of days to weeks in a phenomenon known as representational drift. The tuning stability of individual cells varies over the population, and it remains unclear what drives this heterogeneity. We investigate how a neuron's tuning stability relates to its shared variability with other neurons in the population using two published datasets from posterior parietal cortex and visual cortex. We quantified the contribution of pairwise interactions to behaviour or stimulus encoding by partial information decomposition, which breaks down the mutual information between the pairwise neural activity and the external variable into components uniquely provided by each neuron and by their interactions. Information shared by the two neurons is termed 'redundant', and information requiring knowledge of the state of both neurons is termed 'synergistic'. We found that a neuron's tuning stability is positively correlated with the strength of its average pairwise redundancy with the population. We hypothesize that subpopulations of neurons show greater stability because they are tuned to salient features common across multiple tasks. Regardless of the mechanistic implications of our work, the stability–redundancy relationship may support improved longitudinal neural decoding in technology that has to track population dynamics over time, such as brain–machine interfaces.

**Data availability statement:** The data analysed in this study are from previously published work: Driscoll LN, Pettit NL, Minderer M, Chettih SN, Harvey CD. Dynamic Reorganization of Neuronal Activity Patterns in Parietal Cortex. Cell. 2017;170(5):986–999.e16. doi:10.1016/j.cell.2017.07.021. Data repository: https://datadryad.org/dataset/doi:10.5061/dryad.gqnk98sjq Marks TD, Goard MJ. Stimulus-Dependent Representational Drift in Primary Visual Cortex. Nature Communications. 2021;12(1):5169. doi:10.1038/s41467-021-25436-3. Data repository: https://datadryad.org/dataset/doi:10.25349/D9M606 Code for the analysis performed in this study can be found online: https://github.com/kris-heiney/PID-Drift/.

**Funding:** This project has received funding from the Research Council of Norway (NFR) IKTPLUSS grant (SOCRATES, grant no. 270961, awarded to KH and SN), the NFR FRIPRO grant (DeepCA, grant no. 286558; awarded to SN), the HORIZON EUROPE European Research Council (ERC) starting grant (FLEXNEURO, grant no. 716643, awarded to TO), the Human Frontier Science Program (HFSP) grant (grant no. RGY0069, awarded to TO), the Leverhulme Trust fellowship (awarded to MER), the Isaac Newton Trust fellowship (ECF-2020-352, awarded to MER), the Alexander von Humboldt postdoctoral fellowship (awarded to KH), and the Theoretical Sciences Visiting Program (TSVP) at the Okinawa Institute of Science and Technology (OIST) (awarded to TO). The funders had no role in study design, data collection and analysis, decision to publish, or preparation of the manuscript.

**Competing interests:** The authors have declared that no competing interests exist.

## Author summary

Activity in the brain represents information about the outside world and how we interact with it. Recent evidence shows that these representations slowly change day to day, while memories and learned behaviours stay stable. Individual neurons change their relationship to external variables at different rates, and we explore how interactions with other neurons in the population relates to this neuron-to-neuron variability. We find that more stable neurons tend to share information about external variables with many other neurons. Our results suggest there are constraints on how representations can change over time, and that these constraints are exhibited in shared fluctuations in activity among neurons in the population.

## Introduction

Recent experimental observations of activity in brain regions crucial for processing certain stimuli or behaviours find that populations of neurons change their tuning gradually over days. In a number of cases this occurs in the absence of overt task learning [1–5]. The rate of this 'representational drift' varies from neuron to neuron, with some maintaining stable tuning over days and others showing more volatile responses [6–8].

It is natural to ask what determines the magnitude of drift, and whether features of a neuron's relationship to the task and to other neurons in the population predict the volatility of tuning. Variability in drift rate is related to activity level [7], tuning precision [9], and importance in behavioural decoding [8], but these factors do not entirely explain the population heterogeneity. More recent experimental work has shown that the rate of drift depends on the features of the represented stimulus [4], stimulus familiarity, and the frequency of exposure [2]. Variability in behaviour also contributes to the amount of measured drift in stimulus representation [10]. These findings suggest that a neuron's representations of other variables in the task space contribute to the stability of its tuning to the target variable.

Other work has suggested that a neuron's interactions with the population may drive plasticity in neural representations [11,12]. Sweeney and Clopath [13] have shown that in a network model consisting of neurons with diverse learning rates, faster learners show greater population coupling. In contrast, Sheintuch et al. [9] have shown that in hippocampus, the formation of functionally connected assemblies in CA3 is associated with less drift. The link between population interactions and representational drift thus remains an open question. Furthermore, the reconfiguration of the population code impacts many approaches to studying neural correlates of behaviour, including technologies such as brain–machine interfaces [14]. Being able to predict which couplings—among and between neurons and behaviours—are likely to remain stable would enable an improved capacity to track the neural code as the population drifts.

In this study, we consider how interactions among neurons in a population influence their rates of drift. Why do some neurons in the population exhibit greater tuning stability than others? Specifically, we asked whether the amount of redundancy or synergy, in an information-theoretic sense, is predictive of a neuron's tendency to drift. Intuitively, redundancy measures how much a neuron's firing statistics explain behavioural variability, relative to contribution of other neurons in the population. This relative measure is accompanied by synergy, which quantifies the extent to which pairs of neurons encode complementary aspects of behaviour. Together, these measures account for how behavioural encoding is shared among neurons in a population.

Redundancy and synergy may provide insight into constraints on how population representations can change. Spare degrees of freedom in a mapping between neural activity and behaviour should permit changes in a neuron's tuning curve without entailing a change in behaviour. In this study, we analysed experimental data to evaluate the relationship tuning curve stability has with redundancy and synergy. We found that neurons with high redundant coupling to the population maintain more stable tuning to task variables. Based on our findings, we hypothesise that neurons with high redundant coupling to the population are tuned to a latent task or stimulus feature that is salient across many contexts and thus their activity is more tightly constrained than for other neurons. These neurons may maintain their tuning via the shared connections underlying their redundant activity, providing channels for corrective feedback.

## Results

### Information decomposition as a measure of communication

The phenomenon of representational drift centers on the tuning curve: each neuron's activity is considered in how it relates, on average, to the behaviour or stimulus of interest, and it is said to drift when this relationship changes. However, this perspective neglects communication among neurons in its consideration of only the relationship between the external variable and each single neuron. Furthermore, the tuning curve explicitly disregards trial-to-trial variability by averaging it away. A view of population interactions on a trial-by-trial basis can give more insight into how information is distributed through the population and where each neuron lies in this distributed code.

A common feature of representational drift is that neurons change their tuning in a variety of ways. Our focus in this paper is on possible sources of heterogeneous tuning stability from the perspective of information encoding. We approach this question by reanalysing calcium fluorescence data from experiments by Driscoll et al. [1] and Marks et al. [4], which we briefly describe here.

Schematics of the two experimental setups are shown in Fig 1a. Driscoll et al. [1] recorded from posterior parietal cortex (PPC) daily over weeks as mice navigated a virtual T-maze based on a visual cue given at the start of the maze. Marks et al. [4] recorded primary visual cortex (V1) as mice viewed two types of visual stimulus, passive drifting gratings (PDG) and a movie clip (MOV), weekly over 5–7 weeks. In both experiments, neurons in the population showed heterogeneous stability in their tuning to external variables (Fig 1b). Here we examine how this neuron-to-neuron variability in representational drift relates to information shared among neurons.

In this study, we considered neural representations from an information theoretical perspective. This section gives an overview of what this perspective offers; details of the mathematical formulation and the methods used to apply these calculations to empirical data can be found in the Methods. In an information theoretical context, a neuron's tuning can be characterised by the mutual information $I$ it shares with the external variable. Mutual information quantifies the extent to which knowledge of one variable reduces uncertainty about the state of another variable. The mutual information between a neuron's activity $X$ and an external variable $U$ is given by

$$I(U:X) = \sum_{x \in X} \sum_{u \in U} p(u,x) \log_2 \left[ \frac{p(u,x)}{p(u)p(x)} \right], \tag{1}$$

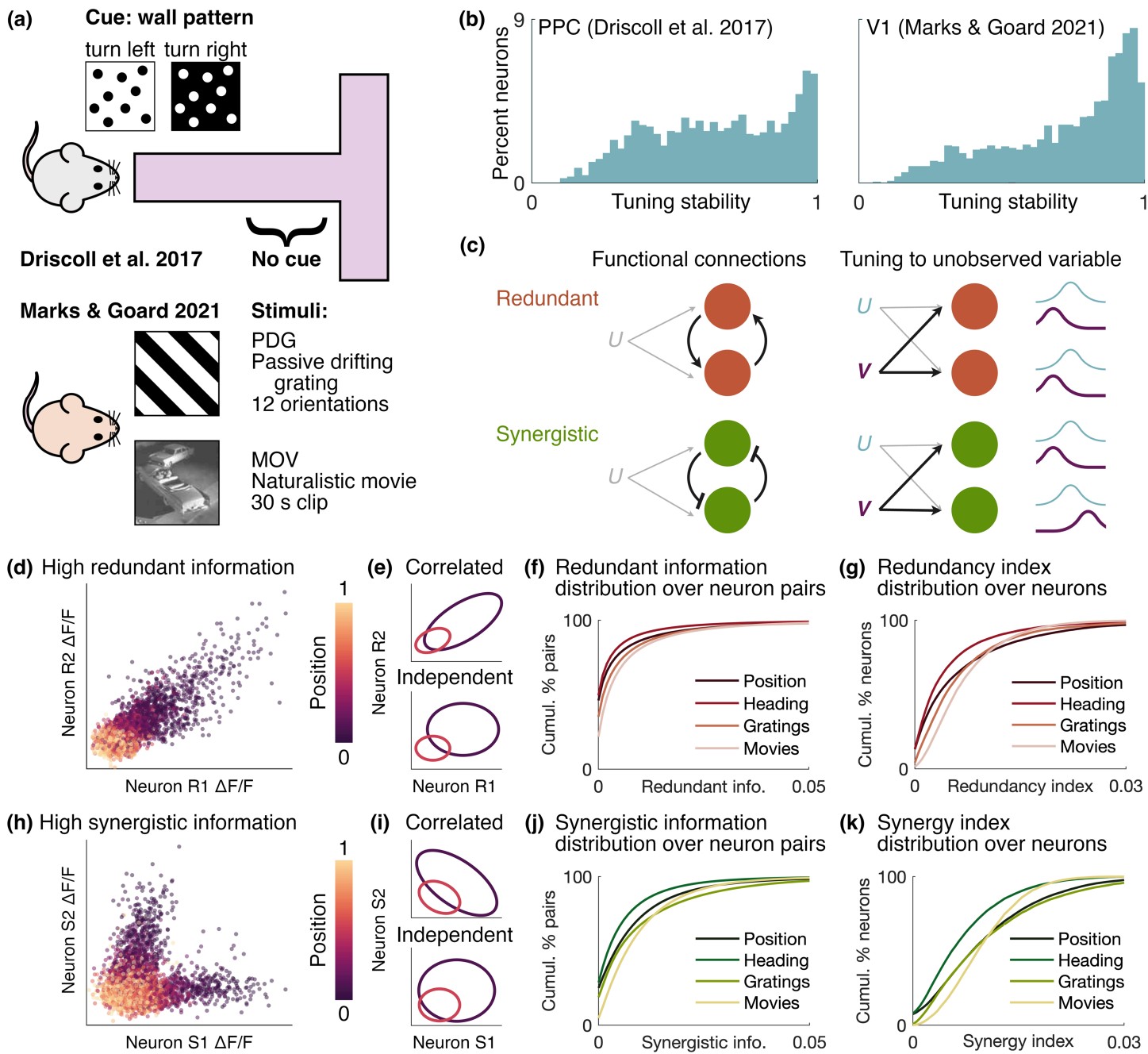

**Fig 1**. **Neural correlations contribute to information encoding of behaviour or stimulus. (a)** We reanalysed data from mouse posterior parietal cortex (PPC) during maze navigation [1] and mouse primary visual cortex (V1) during presentation of artificial (PDG) and naturalistic (MOV) stimuli [4]. **(b)** Histograms of tuning stability (Eq 4) in the two datasets. Note that the stability is computed on different time scales between the two datasets. **(c)** Schematic of potential sources of redundant and synergistic neuron pairs: functional connections between the pair, and tuning to a second unobserved input variable *V*. **(d)** Example PPC neuron pair with high redundancy from mouse 4 on session 5, from [1]. ΔF/F data points are coloured according to the position in the maze. **(e)** Contours of Gaussian distributions fit to the data in **(d)**. We binned the position into 7 bins and fit covariance matrices to bins 1 and 4. The contours in the top plot show these fits, and those in the bottom plot show covariance matrices with the covariance set to 0, corresponding to the neurons acting independent of each other. **(f)** Cumulative distribution of the redundant information over all neuron pairs for each behavior or stimulus. **(g)** Same as **(f)** for redundancy index over all neurons. **(h)–(k)** Same as **(d)–(g)** for synergy.

where $X$ and $U$ are discretised into state bins $x$ and $u$ and $p(\cdot)$ is the probability of a state or joint state occurring. Typically mutual information is considered on a single-neuron basis, but it can be expanded to encompass the information shared between the external variable and a set of multiple neurons. In practice, this presents a statistical challenge, as the number of states that can be taken on by a population increases exponentially with the number of neurons. However, computing the mutual information between the joint activity of a pair of neurons and the external variable is computationally feasible, and it provides insights into how each pair interacts to encode the external variable.

Partial information decomposition (PID) breaks down the mutual information between pairwise neural activity and the external variable into components arising from each neuron independently and from their interactions [15,16,27]. In this view, the joint mutual information for a pair of neurons $\{X_1, X_2\}$ is decomposed into four terms, as

$$I(U : X_1, X_2) = I_{\text{unq}}(U : X_1 \setminus X_2) + I_{\text{unq}}(U : X_2 \setminus X_1) + I_{\text{red}}(U : X_1; X_2) + I_{\text{syn}}(U : X_1; X_2). \qquad (2)$$

The first two terms here are the information $I_{\text{unq}}$ provided uniquely by each neuron independent of the other $\{X_i \setminus X_j\}$, and the latter two relate to the interactions between the neurons $\{X_i; X_j\}$. The third term is the redundant information $I_{\text{red}}$, which is the information shared by the two neurons; in an extreme example where the two neurons were copies of one another, the joint information would be entirely composed of redundant information. The fourth term is the synergistic or complementary information $I_{\text{syn}}$, which is provided only by knowing the state of both neurons together; an example of a synergistic relationship would be an XOR gate, $U = X_1 \oplus X_2$, with $U$, $X_1$, and $X_2$ binary, where the state of $U$ is only known if both $X_1$ and $X_2$ are known.

A number of circuit arrangements can give rise to the statistical signatures of redundancy and synergy (Fig 1c). For example, a pair of neurons with shared tuning to a second external variable $V$ would show high redundancy, as would a pair of neurons that mutually excite one another. In contrast, a pair of neurons with dissimilar tuning to $V$ or a mutually inhibiting pair would show high synergy (see S1 Text for a toy example of synergy).

We applied PID to the two datasets and calculated the decomposition terms in Eq 2 for all pairs of neurons in a population during each experimental session. We computed estimates of these terms using the definitions given by Bertschinger et al. [17] and the empirical methods provided in the IDTxL toolbox by Wollstadt et al. [18] (see Methods).

Example neuron pairs with high redundant (neurons R1 and R2) and synergistic (neurons S1 and S2) information components are shown in Fig 1d and 1h, respectively. These examples are representative of the types of correlations we observe in high-redundancy and -synergy pairs in both datasets. In these plots, the calcium signals ΔF/F of the two neurons are plotted against each other and coloured according to the position of the mouse in the maze. The activity in each of these pairs is strongly correlated but in very different ways: in the high-redundancy pair, when one neuron is strongly activated, the other tends to be as well, whereas in the high-synergy pair, the two neurons are rarely activated together despite both being tuned to a position in the start of the maze. Additional plots showing the tuning curves and redundant and synergistic information of these pairs over sessions can be found in S5 Fig.

Fig 1e and 1i present a schematic view of how activity correlations associated with high redundancy and synergy impact pairwise information about the position in the maze. The more the two ellipses in these plots overlap, the harder it is to distinguish the two position bins they correspond to. In each plot, the two ellipses represent contours of bivariate Gaussian distributions fit to neural activity at two different maze positions, for the same neuron pairs as in Fig 1d and 1h. In the upper plots, the covariance matrices of the Gaussians are taken directly from the data, retaining their position-conditioned correlations. In the lower plots, the diagonal components of the covariance matrices are retained and the off-diagonal components are set to 0: this illustrates the activity if the neurons retained their individual tuning and variance but with no position-conditioned correlation. The effect of setting the covariance to 0 has opposite effects in the redundant and synergistic cases: in the former, there is less overlap than for the nonzero covariance, and in the latter there is more. Although there can be more complex interactions among neurons than these illustrative cases of positive and negative

correlation, these types of correlations are common in the data and provide an intuitive illustration of how redundancy and synergy contribute to the information content of pairwise activity.

Fig 1f and 1j show the cumulative distributions of the redundant and synergistic components of the pairwise information over all neuron pairs for all mice in each dataset, with one distribution shown per behaviour or stimulus. These distributions have a very long tail, with the vast majority of pairs having very low redundant and synergistic information components.

## Redundancy index correlates with tuning stability

We asked whether a neuron's redundant and synergistic interactions with the population are correlated with the stability of the neuron's tuning to an external variable. In the two datasets we studied [1,4], we found that the overall redundant coupling of a neuron to the population generally correlates with tuning curve stability.

We quantified a neuron's degree of redundant and synergistic coupling with the population using the redundancy and synergy indices. For each neuron $X_i$, we calculated and decomposed the joint mutual information between the external variable $U$ and each pair containing that neuron $\{X_i, X_j\}; j \in [1, N], j \neq i$. This gives the four terms in Eq 2 for each pair of neurons containing $X_i$. To obtain the redundancy and synergy indices Red and Syn, we averaged the redundant and synergistic contributions to the joint mutual information over all paired neurons $X_j$, as

$$
\begin{aligned}
\text{Red}_i &= \frac{1}{N-1} \sum_{j=1}^{N} I_{\text{red}}(U : X_i; X_j) \qquad j \neq i, \\
\text{Syn}_i &= \frac{1}{N-1} \sum_{j=1}^{N} I_{\text{syn}}(U : X_i; X_j) \qquad j \neq i.
\end{aligned}
\tag{3}
$$

Fig 1g and 1k shows the cumulative distributions of the redundancy and synergy indices over all neurons in each dataset, for each behaviour and stimulus type. These distributions have long tails, similar to the redundant and synergistic information (Fig 1f and 1j), but with a less drastic drop in probability.

We then evaluated to what extent these indices were correlated with a neuron's tuning stability. As a measure of tuning stability, we used the correlation-based representational drift index (RDI) defined by Marks et al. [4]:

$$
\text{RDI} = \frac{\rho_{ws} - \rho_{bs}}{\rho_{ws} + \rho_{bs}},
\tag{4}
$$

where $\rho_{ws}$ and $\rho_{bs}$ are the within- and between-session cross-correlation, respectively (see Methods).

Neurons with high redundancy or synergy indices tend to be strongly tuned to the target variable. This is because the redundant and synergistic components of the pairwise mutual information are components of how two neurons together encode the stimulus, and when each neuron is taken individually, this information is still present in some form. To ensure any correlation we observed between these PID measures and the tuning stability were not simply due to this strong tuning, we included a measure of tuning strength—the single-neuron mutual information $I(U : X)$ (Eq 1)—as a regressor. We constructed a linear model of stability using a multivariate regression with three regressors: mutual information (MI), redundancy index (Red), and synergy index (Syn) (Fig 2a, further pair plots in S1 Fig). Because of the multi-collinearity of these predictors (Fig 2b), we performed regularised regression with elastic net, which combines L1 and L2 penalties and is useful for constructing models with correlated predictors (see Methods).

Fig 2b and 2c shows the regression coefficients for the three regressors for PPC tuning stability to position in the maze and heading of the animal, and for V1 tuning stability to the drifting gratings and naturalistic movies, respectively. In most cases, the redundancy index was the most strongly weighted predictor of tuning stability. This indicates that the degree

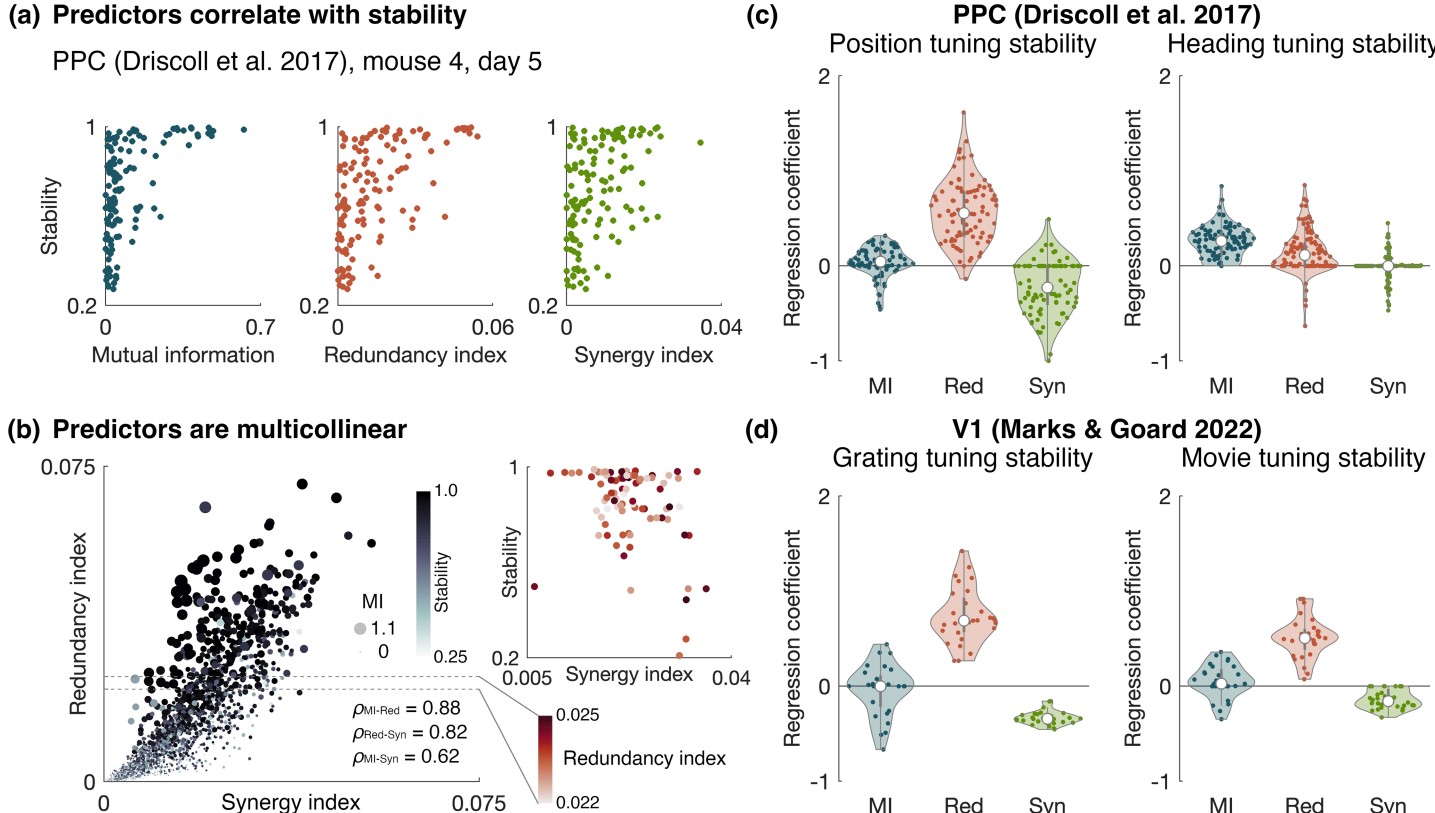

**Fig 2. Redundancy index correlates with tuning stability. (a)** Constructing a multivariate linear model for stability with three regressors: mutual information (MI), redundancy index (Red), and synergy index (Syn). Pair plots showing the stability plotted against each of the predictors for one mouse on a single session (Driscoll et al. [1], 130 neurons, day 5, right turns). **(b)** The three regressors are all strongly correlated with each other. Each data point represents a neuron from one mouse on a given day (Driscoll et al. [1], 130 neurons, 17 days). The redundancy index is plotted against the synergy index, with the size and colour showing the mutual information and stability, respectively. The session-averaged correlation coefficient $\rho$ between each pair of regressors is given in the bottom right. **(c)** Regularised regression coefficients for the three predictors with position and heading as the target external variable [1]. Each data point represents a given session and trial type (left or right turns) for a given mouse. ($n = 617$ PPC neurons across 4 mice.) **(d)** Same as **(c)** but with gratings and movies as the external variable [4]. ($n = 1053$ V1 neurons across 4 mice.)

of redundant coupling to the population is a stronger indicator than the tuning strength of whether a neuron's tuning will remain stable. The coefficient of the mutual information tended to be close to zero, indicating tuning strength provides little predictive power beyond that provided by the redundancy index.

One choice of external variable $U$ did not follow these trends: tuning to the heading of the animal in the PPC. In this case, the mutual information and redundancy index were similarly weighted. This is most likely due to there being a sparser representation of heading among the recorded neurons. A pair of neurons must both be responsive to similar values of $U$ to have any appreciable redundant or synergistic information components. Therefore, with a sparse representation, there is less chance for overlap in neurons' tuning to $U$, making redundancy and synergy generally low across the population and therefore less predictive of stability. However, this result suggests that the relationship between stability and redundancy index is not a statistical artifact.

Interestingly, the coefficient for the synergy index tended to be negative. This means that independent of the other variables, the synergy index has a negative relationship with the tuning stability. We demonstrate this in a plot the tuning stability is against the synergy index for neurons within a narrow range of redundancy indices (inset of Fig 2a). Each data

point in this plot corresponds to a neuron with a redundancy index in the target range on a given day and is colored according to redundancy index. As shown here, although the stability generally exhibits a positive trend with the synergy index, when the redundancy index is fixed, an underlying negative trend emerges.

To better understand this interplay among redundancy, synergy, and stability, we recast these network features as graphs whose adjacency matrices are given by the redundant and synergistic information for each pair of neurons (Fig 3a). In these graphs, the strongest redundant edges tended to be confined to the subset of the population with the highest stability, further supporting the relationship between redundancy and stability.

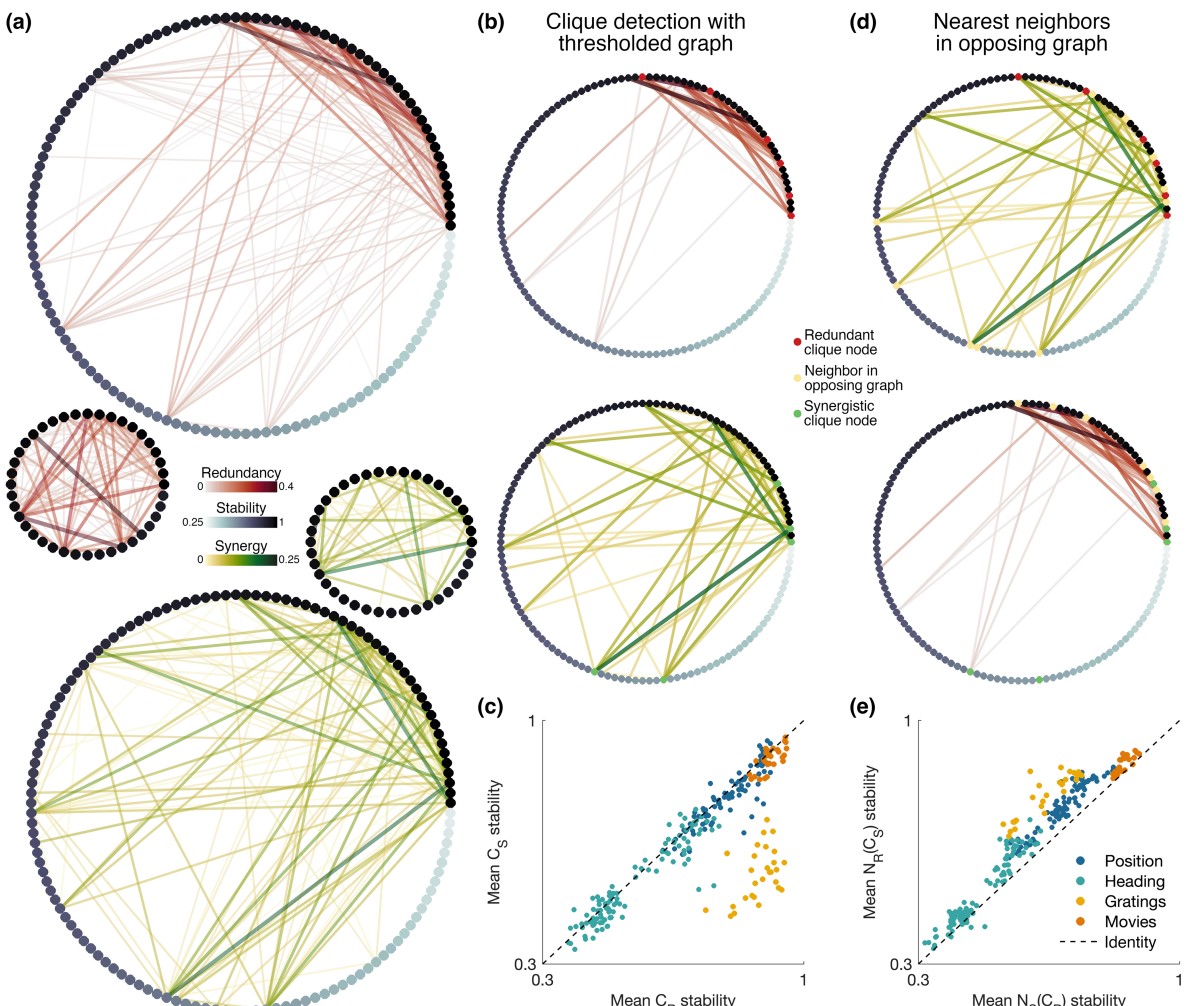

**Fig 3**. **High-stability neurons form redundant cliques. (a)** Graph plots of pairwise redundant and synergistic information. Nodes represent neurons and are coloured and ordered according to position tuning stability. Edges represent the redundant or synergistic component of the pairwise information, thresholded for the purposes of visualisation. Inset graphs show the top 40 most stable neurons in the population. **(b)** Clique detection in thresholded graphs $R$ and $S$ retaining the top 0.35% of edges. Redundant and synergistic cliques $C_R$ and $C_S$ are highlighted in red and green, respectively. **(c)** Mean stability of redundant and synergistic cliques $C_R$ and $C_S$. Each data point is an average over 10–50 size-matched graph pairs for a single session and mouse (see Methods). **(d)** Nearest neighbors $N_S(C_R)$ and $N_R(C_S)$ of redundant and synergistic cliques $C_R$ and $C_S$ in their opposing graphs. Nearest neighbors are highlighted in yellow. **(e)** Mean stability of opposing nearest neighbors $N_S(C_R)$ and $N_R(C_S)$. Each data point is an average over 10–50 size-matched graph pairs for a single session and mouse (see Methods). Illustrative graph plots in **(a)**, **(b)**, and **(d)** are from a single session for one mouse (Driscoll et al. [1], 130 neurons, day 4, left turns).

The structure of these graphs suggests that the most stable neurons in the population tend to be strongly redundantly interconnected. To quantify this, we thresholded redundancy and synergy graphs over a range of thresholds and binarised the resulting graphs (see Methods). We then detected the largest fully connected subgraphs or 'cliques' $C_R$ and $C_S$ in the binarised graphs $R$ and $S$ (illustrated in Fig 3b). To demonstrate the tendency of high-stability neurons to form redundant cliques, we compared the mean clique stability in size-matched $R$–$S$ graph pairs (i.e. graphs whose largest cliques $C_R$ and $C_S$ containing the same number of nodes, Fig 3c). In this way, the synergistic graph acts as a sort of null model. To evaluate the stability of redundant cliques, we considered the difference in the mean stability of size-matched cliques $C_R$ and $C_S$, $\lambda_C = \bar{\lambda}(C_R) - \bar{\lambda}(C_S)$. The stability of $C_R$ tended to be higher than that of $C_S$, and a difference of $\lambda_C = 0$ lay outside of the 95% bootstrap confidence interval for nearly all mice and choices of external variable $U$, except in three instances (of $n = 24$ total combinations of mouse and $U$; S4a and S4b Fig).

These high-stability redundant cliques tended to connect synergistically to the rest of the population. To evaluate this tendency, we considered the mean stability of the nearest synergistic neighbors $N_S(C_R)$ of the redundant cliques $C_R$ (illustrated in Fig 3d). As with the mean clique stability, we used the redundant neighbors $N_R(C_S)$ of the synergistic cliques $C_S$ as a null model for comparison (Fig 3e) and computed the difference in nearest neighbor stability, $\lambda_N = \bar{\lambda}(N_S(C_R)) - \bar{\lambda}(N_R(C_S))$, for size-matched clique pairs. The synergistic neighbors of the redundant cliques exhibited lower stability than the converse, and a difference of $\lambda_N = 0$ lay outside of the 95% bootstrap confidence interval for all mice and choices of external variable $U$ (Fig S4c and S4d). Thus, redundant cliques in the network exhibit high stability and connect synergistically to the remaining lower-stability portion of the population.

## Discussion

The effects of representational drift are not seen uniformly across neurons in a population, and what contributes to this heterogeneity remains an open question. Although the results of Sweeney and Clopath [13] suggest neurons with more plastic representations couple more strongly to the population, our analysis is more aligned with other studies [9,19] showing that neurons participating in functional ensembles tend to show greater tuning stability.

Here, we focused specifically on the effect of stimulus-conditioned correlations on information encoding. Our results demonstrate that the most stable cells in a population tend to participate in redundantly connected cliques that in turn synergistically connect to the rest of the population, complementing previous work on how the structure of interaction information graphs influences network computation [20,21]. Additionally, high synergy tends to contribute negatively to the tuning stability. One manifestation of this is that among the highly redundant subpopulations in PPC, those neurons with greater synergistic coupling to the population tend to be less stable (inset of Fig 2a).

Redundant and synergistic coupling may arise from either structural connections or tuning relationships among other unobserved variables (Fig 1c), and it is challenging to determine which of these sources contributes to the redundancy we observed in this study. A closer look at select pairs of neurons gives an indication that both of these sources are present in the current analysis. In some neuron pairs with high redundant information, we observed small corresponding changes in tuning (see tuning curves of example neuron pair in S5b Fig). In other pairs, redundancy with respect to maze position is likely attributable to tuning to heading (see S6 Fig), though the correlation among the task variables in this experiment makes it challenging to determine which variable or combination of variables the neurons respond to. In either case, the relationships underlying redundant coupling are indicative of a more complex population tuning to the task space not accounted for when considering a single neuron's tuning to a single external variable [22].

On the basis of these observations, we conjecture that redundancy may reveal how drift is constrained across the population in a way that limits interference with relationships among task variables that are more generally relevant across many tasks. In the case that a neuron pair is structurally connected, their connection may serve a reinforcing effect, essentially making it harder for their tuning to be disturbed by outside fluctuations to only one neuron of the pair. In the case of co-tuning to an unobserved variable, neuron pairs being similarly responsive to not only the target task variable

but also a second unobserved variable indicates this combination of task variables recruits more neurons to represent it, which is suggestive of high salience.

In contrast to the case of redundancy, high-synergy pairs arising from structural connections would likely involve mutual inhibition. It is possible that such pairings would produce stimulus-specific competition, driving more changes in tuning than would occur without such interactions. It is widely considered that synergy is a advantageous approach to information encoding in the brain, as it takes advantage of the full range of combinations of activations in the system; however, it is also fragile and susceptible to disruption by a small number of failures in the network [23]. This suggests that while it would be valuable to exploit synergistic interactions in the neural code, it may also be a good strategy for the brain to allow some flexibility in reading out information from neurons with many such interactions.

Our observations here are somewhat limited by our discretisation of the task space and the confounding effect of the fixed relationships among the task variables. Further analysis using a different experimental paradigm and computational models would be beneficial to provide greater support for our findings. Despite these limitations, this work suggests the possibility that stable subsets of the population encode for generally salient task features, evident in the stimulus-conditioned relationships among the neuronal responses. Further experimental work in which the relationships among task variables could be controlled would help clarify the computational role of highly redundant neurons. Regardless of the mechanistic interpretation of the redundancy–stability relationship, our results also present a step toward a practical approach to predicting which neurons will maintain their relationship with the external variable, which is useful in technology reliant on long-term neural decoding such as brain–machine interfaces [14].

## Conclusion

In this study, we evaluated the relationship among redundancy, synergy, and tuning curve stability in two longitudinal neural datasets. We found that neurons with higher redundant coupling to the population tend to exhibit higher tuning stability. Neurons tend to organise into highly redundant and highly stable cliques that in turn synergistically connect to the remainder of the population. We also found that synergy tends to be negatively correlated with tuning stability. We theorise that neurons with high redundant coupling to the population may be tuned to highly salient aspects of the task, motivating their more stable relationship to task variables, whereas neurons with high synergistic coupling may be driven to drift more as a result of competition. Regardless of the mechanism underlying our results, our finding can benefit work where the relationships between neural activity and task variables must be tracked over long timespans, such as brain–machine interface research.

## Methods

### Experimental data

The two-photon calcium imaging and behavioural data used in this study are from Driscoll et al. [1] and Marks et al. [4], and complete details of the experiments and data can be found there.

**Posterior parietal cortex.** In Driscoll et al. [1], mice were trained over a 4–8 week period to perform a two-alternative forced-choice task in a virtual environment: the mice were given a visual cue at the start of a T-maze and had to turn left or right when they reached the T-intersection. Mice received a reward for correct associations of visual cues with left or right turns. The data are available from the Dryad repository [24].

The MATLAB-based software ViRMEn (Virtual Reality Mouse Engine) [25] was used to construct the virtual environment and collect behavioural data. Driscoll et al. [1] collected calcium fluorescence imaging data at a sampling rate of 5.3 Hz and identified and segmented fluorescence sources. The processed fluorescence data used here consist of normalised fluorescence traces ($\Delta F/F$).

In the present analysis, we excluded data from the inter-trial intervals and incorrect trials, and we analysed left- and right-turn trials separately, as many neurons display trial-type-specific activity [1]. Thus, the neuron response signal $x(t)$

represents $\Delta F/F$ at time sample $t$, where $t$ includes all time samples of all correct trials of a single type (i.e., left or right turn) on a given session. We considered two behavioural signals: position along the T-maze and heading angle.

Neurons had to be present (confidence index of 1 or 2) in at least 80% of the considered sessions (see Sect ) to be included in the analysis. We included five populations of neurons from four mice: mouse 3, 194 cells, 10 sessions (sessions 1, 2, 4, 6–12); mouse 3, 177 cells, 10 sessions (sessions 13–22); mouse 4, 130 cells, 17 sessions (sessions 1–17); mouse 5, 130 cells, 7 sessions (sessions 7–13).

**Primary visual cortex.** In Marks et al. [4], mice passively viewed two different types of visual stimuli: oriented gratings and a naturalistic movie. The gratings consisted of 12 different orientations evenly spaced from 0° to 330° in 30° increments, presented in 8 trials. A single trial contained each orientation presented for 2 s followed by a 4-s inter-stimulus interval of a grey screen, with all orientations presented in ascending order of orientation angle. The naturalistic movie was a continuous 30-s clip from the film *Touch of Evil*, presented in 30 trials with a 5-s inter-trial interval between each presentation (grey screen). Mice viewed these stimuli once per week for 5–7 weeks; from the full dataset, we selected mice who had sessions for 7 weeks (mice 2, 10, 11, and 12 from the main experiment, mice 1–4 from the inhibitory experiment). The data are available from the Dryad repository [26].

Marks et al. [4] collected calcium fluorescence imaging data at a sampling rate of 10 Hz and identified and segemented fluorescence sources. As with the data from Driscoll et al. [1], we performed our analysis on the normalized fluorescence traces ($\Delta F/F$).

We excluded data from the inter-trial and inter-stimulus intervals, and we analysed the grating and movie trials separately. We constructed stimulus variables as the orientation angle of the gratings and the time elapsed in the movie.

As above, neurons had to be present (confidence index of 3) in at least 80% of sessions to be included in the analysis. We included four populations of excitatory neurons from four mice in the main experiment: mouse 2, 355 neurons; mouse 10, 252 neurons; mouse 11, 227 neurons; mouse 12, 219 neurons. We also analysed four populations of inhibitory neurons from four mice in the inhibitory experiment: mouse 1, 66 neurons; mouse 2, 76 neurons; mouse 3, 70 neurons; mouse 4, 48 neurons.

## Data discretisation

The nonparametric, information-theoretic estimators (mutual information, $\Delta I$) used in this work require histogram-based estimators of various joint densities. To make this problem tractable, we discretised the fluorescence and behavioural signals prior to analysing the data (see [27] for details on data binning methods). Limitations of the data indicated a coarse bin size. We verified our binning approach was sufficient to support our conclusions by binning at multiple resolutions (details below).

We binned the fluorescence signals $X(t) = \{\frac{\Delta F}{F}\}(t)$ from each neuron into three unform-width bins. To reduce the influence of outliers, we first determined the $5^{th}$–$95^{th}$ percentile range $\{a, b\} = \{\text{percentile}[X, 5], \text{percentile}[X, 95]\}$. We then defined the bin edges as $\{\min[X], a + \frac{1}{3}(b - a), a + \frac{2}{3}(b - a), \max[X]\}$. That is, we binned data within the $5^{th}$–$95^{th}$ percentile range into three equally spaced bins, and assigned data below the $5^{th}$ or above the $95^{th}$ percentiles (saturation bounds) into the lower and upper bin, respectively.

The grating orientation stimulus variable from the Marks et al. [4] dataset is already a set of 12 discrete angles (0° to 330° in intervals of 30°) and thus requires no further binning. We binned the movie stimulus into 12 temporal bins of 2.5 s each.

We binned the behavioural signals from the Driscoll et al. [1] dataset (position and heading) into $N = 5, 10, 20$ state bins using two methods: uniform-width binning for quantifying tuning stability and uniform-activity binning for information theoretic metrics. Similar results were obtained for 10 and 20 bins.

We binned the heading into uniform-width bins in the same manner as with the fluorescence: with saturation bounds at the upper and lower fifth percentiles to account for extreme values. Because the position signal spans the T-maze

track and is not influenced by extreme values in the same way as the other behaviours, we used the full range of position values to define the limits $(a,b)$ for uniform-width binning. We used these uniform-width binned signals to calculate the change in tuning curves between a pair of days; this choice was made to ensure the bin edges were consistent across days.

In the Driscoll et al. [1] data set, neural population tuning is not distributed uniformly over all behavioural states when states are defined using the uniform-width approach. For example, more neurons respond near the beginning and end of the maze, and so the first and last position bins would be accompanied by higher population activity. For this reason, previous studies [27] have suggested partitioning covariates into bins containing (approximately) equal amounts of spiking activity. This results in higher-resolution binning for regions in behavior space that drive population activity more strongly.

In these data, we do not have direct access to spike counts, so we use the population-average $\Delta F/F$ as a proxy for this uniform-activity binning. For each behavioural variable $U(t)$, we adaptively selected bin edges $\text{bins}[U(t)]$ such that the integral of population-fluorescence activity was the same for each bin. That is, we (i) calculated the average total population activity for a given behaviour state $u$ (over all correct left- or right-turn trials within a session), and (ii) adjusted the bin edges so that the sum of this average activity in each bin equaled the same constant $C$. Formally, this can be written as:

$$\text{bins}[U(t)] = \{b_0, ..., b_N\}, \text{ such that } \int_{b_{i-1}}^{b_i} \bar{\boldsymbol{X}}(U')\,\mathrm{d}U' = C \text{ for } i \in \{1..N\}, \tag{5}$$

where $\bar{\boldsymbol{X}}(U')$ is the average population $\Delta F/F$ associated with the specific value $U'$ of the behavioral covariate. This is given by

$$\bar{\boldsymbol{X}}(U') := \frac{\int \|\boldsymbol{X}(t)\|_1 \delta(U(t) - U')\,\mathrm{d}t}{\int \delta(U(t) - U')\,\mathrm{d}t}, \tag{6}$$

where $\|\boldsymbol{X}(t)\|_1$ is the sum of the population fluorescence activity for time-point $t$, and $\delta$ are Dirac deltas indicating times when the behavioral covariate $U(t)$ has the specified value $U'$. We used the resulting uniform-activity binned signals to calculate the mutual information.

## Mutual information

We used the mutual information to quantify the strength of a neuron's tuning to a target behavioural variable. The mutual information $I(U:X)$, in bits, between a neuron response $X(t)$ and a behaviour $U(t)$ is given by

$$I(U:X) = \sum_{u \in U} \sum_{x \in X} p(u, x) \log_2\left[\frac{p(u, x)}{p(u)p(x)}\right], \tag{7}$$

where $u$ and $x$ are the state values of the discretised behaviour $U$ and response $X$ vectors, respectively, and $p(\cdot)$ denotes the probability.

We used a shuffle test to determine whether each neuron was significantly tuned to the target behaviour. For this test, we shuffled the time samples of the concatenated within-trial neuron responses 1000 times for each neuron and recalculated the mutual information between the behaviour and the shuffled signals. If the mutual information of the unshuffled signal is greater than 95% of the shuffled cases, the neuron is considered tuned to the behaviour.

## Quantifying synergy and redundancy

To quantify how stimulus-conditioned correlations between pairs of neurons influence the information the pair encodes about an external variable $U$, we decomposed the pairwise mutual information into its constituent components. The mutual

information $I(U : X_1, X_2)$ between an external variable $U$ and a pair of neurons $\{X_1, X_2\}$ can be decomposed into four terms, as

$$I(U : X_1, X_2) = I_{\text{unq}}(U : X_1 \setminus X_2) + I_{\text{unq}}(U : X_2 \setminus X_1) + I_{\text{red}}(U : X_1; X_2) + I_{\text{syn}}(U : X_1; X_2), \tag{8}$$

where $I_{\text{unq}}$ is the information provided uniquely by one of the neurons (unique information), $I_{\text{red}}$ is the information both neurons share about $U$ (redundant information), and $I_{\text{syn}}$ is the information they provide together (synergistic information).

Various methods of estimating the terms in Eq 8 have been proposed. In this study, we used the formulation by Bertschinger et al. [17] and the empirical methods to compute them provided in the IDTxL toolbox by Wollstadt et al. [18]. Following Bertschinger et al. [17], the four components of the information can be estimated from data as

$$\tilde{I}_{\text{unq}}(U : X_i \setminus X_j) = \min_{Q \in \Delta_P} I_Q(U : X_i | X_j)$$
$$\tilde{I}_{\text{red}}(U : X_i; X_j) = \max_{Q \in \Delta_P} \left[ I_Q(U : X_i) - I_Q(U : X_i | X_j) \right] \tag{9}$$
$$\tilde{I}_{\text{syn}}(U : X_i; X_j) = I(U : X_i, X_j) - \min_{Q \in \Delta_P} I_Q(U : X_i, X_j).$$

In the notation used in Eqs. 8 and 9, a colon between variables indicates the mutual information is computed between them, a backslash indicates information present in the former variable but not the latter (unique information), and a semi-colon indicates information related to the interaction of the two variables (redundant or synergistic information).

Eq 9 involves maximising or minimising information terms $I_Q$ across the probability distributions $Q$ in the set of distributions $\Delta_P$. Letting $\Delta$ be the set of all joint distributions of the three variables of interest $\{U, X_i, X_j\}$, $\Delta_P$ is the subset of $\Delta$ in which the marginal distributions on the pairs $(U, X_i)$ and $(U, X_j)$ are the same as in the empirical distribution $P$:

$$\Delta_P = \Big\{ Q \in \Delta : Q(U = u, X_i = x_i) = P(U = u, X_i = x_i)$$
$$\text{and} \ \ Q(U = u, X_j = x_j) = P(U = u, X_j = x_j) \Big\}.$$

As in the definition of the mutual information, capital letters $U$ and $X$ represent state values of the discretised signals $u$ and $x$. We performed the optimisations in Eq 9 using the method by Makkeh et al. [28], as implemented in IDTxL [18]. To reduce the influence of bias on the PID measures, we performed shuffle-subtraction for bias correction [29]. We recomputed the information terms with the task variable shuffled and subtracted the resultant shuffle information from the estimates.

To characterise the tendency of a neuron to interact redundantly and synergistically with the population, we defined redundancy and synergy indices for each neuron. The redundancy and synergy indices Red and Syn are obtained by averaging the redundant and synergistic contributions to the joint mutual information over all paired neurons $X_j$, as

$$\text{Red}_i = \frac{1}{N-1} \sum_{j=1}^{N} I_{\text{red}}(U : X_i; X_j) \qquad j \neq i,$$
$$\tag{10}$$
$$\text{Syn}_i = \frac{1}{N-1} \sum_{j=1}^{N} I_{\text{syn}}(U : X_i; X_j) \qquad j \neq i,$$

where $N$ is the number of neurons in the population.

## Quantifying tuning stability

We quantified the tuning stability of each neuron by comparing its tuning curves between pairs of sessions. To evaluate the robustness of the observed correlations to methodological choices, we used three tuning curve comparison metrics— (1) the cross-correlation [4], (2) the cosine similarity [30], and (3) the mean of absolute differences. The main results are reported using the cross-correlation–based method, for the sake of consistency with the original results from Marks et al. [4]. However, all three stability methods yielded results consistent with those shown in Fig 2. Each of the three methods are described in more detail below.

First, we normalised the tuning curves as follows. The raw session-wise tuning curves are given as an $N$-dimensional vector, where $N$ is the number of behavioural bins, and each component in the vector is the average $\Delta F/F$ across all samples in the given behavioural bin. These vectors are then z-scored to give the normalised tuning curve.

The cross-correlation metric of stability is given by

$$
\begin{aligned}
\lambda_{\mathrm{cc}}(m, n) &= \frac{\rho_{ws} - \rho_{bs}}{\rho_{ws} + \rho_{bs}} \\
&= \frac{(\rho_{m,m} + \rho_{n,n}) - 2\rho_{m,n}}{(\rho_{m,m} + \rho_{n,n}) + 2\rho_{m,n}}
\end{aligned}
\tag{11}
$$

where $\rho_{ws} = \frac{1}{2}(\rho_{m,m} + \rho_{n,n})$ and $\rho_{bs} = \rho_{m,n}$ are the Pearson correlations within and between sessions. To calculate the within-session correlation, the trials are randomly partitioned in half and the correlation is taken between the two halves. This partitioning is performed 1000 times and the final within-session correlation is the average over the 1000 partitions. The overall tuning stability for the neuron is then given by $\Lambda_{cc} = \langle \lambda(m, n) \rangle_{m,n; \, m \neq n}$, the average over all pairs of sessions, as is also the case for the other two metrics.

The cosine similarity metric of stability [8] is given by

$$
\lambda_{\cos}(m, n) = \frac{\mathbf{z}^m \cdot \mathbf{z}^n}{||\mathbf{z}^m|| \, ||\mathbf{z}^n||},
\tag{12}
$$

where $\mathbf{z}^m$ and $\mathbf{z}^n$ are the normalised tuning curves on sessions $m$ and $n$, respectively. This metric is then normalised by the within-session cosine similarity during each session $m$ and $n$. As with the cross-correlation metric, the within-session stability is averaged over 1000 random partitions. This approach considers the 'alignment' of the tuning curve vectors in $N$-dimensional space and can range from 0 (orthogonal) to 1 (perfectly aligned).

For the mean of absolute differences method, the tuning stability $\lambda(m, n)$ for a pair of sessions $m$ and $n$ ($m \neq n$) is given by

$$
\lambda_{\mathrm{diff}}(m, n) = 1.2 - \left\langle \mathbf{z}_i^m - \mathbf{z}_i^n \right\rangle_i,
\tag{13}
$$

where $\mathbf{z}_i^m$ and $\mathbf{z}_i^n$ are the $i$th components of the normalised tuning curves on sessions $m$ and $n$, respectively. The subtraction from 1.2 is included to yield stability values that are increasingly positive for increasingly similar tuning curve pairs; the selection of the constant value was chosen to yield stability values generally falling in the range of 0 to 1.

## Statistical relevance of predictors of drift

A primary goal of this paper was to determine whether redundancy and synergy are predictive of a neuron's stability. To this end, we performed regression to fit the following model:

$$
y = \beta_0 + \beta_1 x_1 + \beta_2 x_2 + \beta_3 x_3,
\tag{14}
$$

where the response variable $y$ is the tuning curve stability and the predictors $x_i$ are the mutual information, redundancy index, and synergy index with corresponding regression coefficients $\beta_i$. Here, the response variable represents the tuning stability averaged over all pairs of sessions, but each of the predictors are taken as single-session values with regressions performed separately for each session. The three predictor variables and the response variable were all z-scored before the regression analyses.

We used regularised regression, specifically elastic net (MATLAB function `lasso`), to evaluate which predictor variables contribute to describing variance in stability and rank their predictive relevance [31]. This regularisation method is useful when handling linear models with strongly multicollinear predictors, as it penalises high coefficients to avoid overfitting. The resulting coefficients can thus be used to rank the relevance of the corresponding predictors in the model.

Elastic net combines the L1 penalty of lasso regression with the L2 penalty of ridge regression, and we used a mixing parameter of $\alpha = 0.5$ to define the loss function. The selection of the $\alpha$ parameter did not qualitatively change the regression results. We selected the $\lambda$ parameter corresponding to the model with the lowest 10-fold cross-validated mean squared error to obtain the final coefficient values. If any predictor did not have a significant correlation with the stability in a single-variable model (i.e., $p > 0.05$), its regression coefficient was set to 0.

To consider the effect of variable interactions, we performed an additional regression analysis including interaction terms:

$$y = \beta_0 + \beta_1 x_1 + \beta_2 x_2 + \beta_3 x_3 + \beta_{12} x_1 x_2 + \beta_{13} x_1 x_3 + \beta_{23} x_2 x_3. \tag{15}$$

The results with the interaction terms are shown in S2 and S3 Figs.

**Clique detection and graph analysis**

For each experimental session, we generated two weighted graphs describing the pairwise information components among the neurons (nodes) in the population, one graph with edges corresponding to the redundant information, and one corresponding to the synergistic information (see Fig 3a). We detected cliques in each of these graphs and their nearest neighbours in the opposite size-matched graph using the following methods.

Starting with the weighted graphs, we first systematically thresholded the edges and binarised the thresholded graphs to produce unweighted redundant and synergistic graphs $R$ and $S$. We set thresholds to correspond to the inclusion of the top $n$ percent of edges in the graph, with $n \in (0.25, 0.50, ...n_{max})$. For the PPC and V1 datasets, the maximum percentages $n_{max}$ were 25% and 15%; these were set in consideration of computation time, and the V1 threshold was lower because of the greater number of neurons in this dataset. We then detected all maximal cliques in the unweighted graphs using the method by Eppstein et al. [32], as implemented in the function `ELSclique` [33]. We defined the size of a clique as the number of nodes it contains and the largest cliques as the cliques containing the most nodes. S4a Fig shows an example pair of thresholded redundant and synergistic graphs $R$ and $S$ from the PPC dataset. These graphs include the top 0.25% highest edges. The nodes in the largest redundant and synergistic cliques $C_R$ and $C_S$ are coloured red and green, respectively.

For each session, we then size-matched the resultant set of largest cliques between the redundant and synergistic graphs. That is, we defined a pair of thresholded redundant and synergistic graphs as size-matched when their largest cliques were the same size. When multiple considered thresholds yielded the same largest clique size, we selected the graph with the fewest edges. S1 Table gives the range of clique sizes obtained with the considered thresholds and size-matching approach, along with the numbers of size-matched $R$–$S$ graph pairs.

We compared the stability of all size-matched pairs of redundant and synergistic cliques using bootstrapping. We first calculated the difference in mean clique stability $\lambda_C = \bar{\lambda}(C_R) - \bar{\lambda}(C_S)$ for all size-matched $R$–$S$ graph pairs from each mouse. We then generated 1,000 bootstrapped replicates of this differential stability $\lambda_C$ by sampling with replacement. Histograms and confidence intervals of the means of 1,000 bootstrapped replicates of $\lambda_C$ are plotted in S4a and S4b Fig.

We also determined the nearest neighbours $N_S(C_R)$ and $N_R(C_S)$ of the cliques $C_R$ and $C_S$ in their opposing graphs $S$ and $R$, respectively, as illustrated in Fig 3d. We compared the stability of these nearest neighbours in size-matched graph pairs again using bootstrapping. As with the clique stability, we calculated the difference in mean neighbor stability $\lambda_N = \bar{\lambda}(N_S(C_R)) - \bar{\lambda}(N_R(C_S))$ for all size-matched $R$–$S$ graph pairs from each mouse. Histograms and confidence intervals of the means of 1,000 bootstrapped replicates of $\lambda_N$ are plotted in S4c and S4d Fig.

## Supporting information

**S1 Fig. Pairwise plots of predictors and response variable.** Example pairwise plots of considered variables for one mouse (mouse 4, PPC, external variable: position, left-turn trials) on different sessions. Each data point represents one neuron ($n = 194$ neurons). Rows show a selection of six sessions (sessions 1, 4, 7, 10, 13, and 16) out of a total of 17 sessions. Columns show a given pairwise plotting. The four plotted variables are the tuning stability, mutual information, redundancy index, and synergy index. In the regressions performed in this study, these are respectively the response variable and the three predictors.
(TIFF)

**S2 Fig. Regression coefficients obtained by different regression methods.** Comparison of regression coefficients obtained by elastic net **(a,b)** without and **(c,d)** with interaction terms. In the regressions, the response variable was the tuning stability, and the three predictors were mutual information (MI), redundancy index (Red), and synergy index (Syn), with maze position as the target external variable. Each data point plotted here represents regression coefficients for one mouse on a single session. **(a,c)** Regression coefficients plotted per neural population: mouse 3, 194 cells, 10 sessions (sessions 1, 2, 4, 6–12); mouse 3, 177 cells, 10 sessions (sessions 13–22); mouse 4, 130 cells, 17 sessions (sessions 1–17); mouse 5, 130 cells, 7 sessions (sessions 7–13). **(b,d)** Regression coefficients across all mice. Elastic net was performed using MATLAB's `lasso` function.
(TIFF)

**S3 Fig. Stepwise regression coefficients including interaction terms for all task variables.** Regression results are shown for each considered external variable: **(a)** position, **(b)** heading, **(c)** gratings, and **(d)** movies.
(TIFF)

**S4 Fig. Redundant cliques have high stability and their immediate synergistic neighbours have low stability.** Histograms and 95% confidence intervals of means from 1,000 dataset replicants generated by bootstrap resampling with replacement. **(a)** Difference in mean stability of size-matched redundant and synergistic cliques, $\lambda_C = \bar{\lambda}(C_R) - \bar{\lambda}(C_S)$. Histograms are pooled over mice. **(b)** Mean and 95% confidence intervals per mouse. For $U$ the position or heading, left- and right-turn trials are plotted separately. **(c,d)** Same as **(a,b)** for difference in mean stability of clique nearest neighbors in opposing graph, $\lambda_C = \bar{\lambda}(C_R) - \bar{\lambda}(C_S)$.
(TIFF)

**S1 Table. Graph analysis parameters for S4 Fig.** Parameters of graphs used to compare clique $\{C_R, C_S\}$ stability and clique nearest neighbour $\{N_S(C_R), N_R(C_S)\}$ stability in S4 Fig. See Methods for details. Clique size is given as a range of percentages of the total number of cells. The number of $R$–$S$ pairs gives the range of the per-session number of paired unweighted $R$ and $S$ graphs with equal clique sizes.
(PDF)

**S5 Fig. Illustrative examples of neuron pairs with coordinated variability typical of (a)–(c) high redundant information and (d)–(f) high synergistic information.** These pairs correspond to those shown in Fig 1 (PPC dataset, external variable: position, mouse 4). **(a)** Neural activity of a highly redundant neuron pair, Neurons R1 and R2, on two different

sessions. ΔF/F samples are colored according to the normalised position in the maze. **(b)** Tuning curves of the two neurons in **(a)**, with sessions stacked vertically. **(c)** Redundant and synergistic information for the pair shown in **(a)** plotted over sessions. **(d)–(f)** Same as **(a)–(c)** for a pair with high synergistic information, Neurons S1 and S2.
(TIFF)

**S6 Fig. Redundancy can arise from co-tuning to an unmeasured variable. (a)–(c)** Same as S5a-c Fig for a neuron pair with high redundancy, Neurons H1 and H2. **(d)–(f)** Same as **(a)–(c)** but with heading, instead of position, as the external variable.
(TIFF)

**S1 Text. Toy example of synergy.**
(PDF)

## Acknowledgments

We would like to thank Joseph Lizier and Michael Goard for insightful discussions about this work.

## Author contributions

**Conceptualization:** Kristine Heiney, Mónika Józsa, Timothy O'Leary.

**Formal analysis:** Kristine Heiney, Mónika Józsa, Michael E. Rule.

**Funding acquisition:** Kristine Heiney, Stefano Nichele.

**Investigation:** Kristine Heiney, Mónika Józsa, Timothy O'Leary.

**Methodology:** Kristine Heiney, Mónika Józsa, Michael E. Rule, Timothy O'Leary.

**Project administration:** Stefano Nichele, Timothy O'Leary.

**Resources:** Henning Sprekeler, Stefano Nichele, Timothy O'Leary.

**Supervision:** Mónika Józsa, Michael E. Rule, Henning Sprekeler, Timothy O'Leary.

**Validation:** Kristine Heiney.

**Visualization:** Kristine Heiney, Mónika Józsa.

**Writing – original draft:** Kristine Heiney, Mónika Józsa, Timothy O'Leary.

**Writing – review & editing:** Kristine Heiney, Mónika Józsa, Michael E. Rule, Henning Sprekeler, Stefano Nichele, Timothy O'Leary.

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
