## [Decision Letter · Decision Letter 0]

12 Jun 2025

PCOMPBIOL-D-25-00928

Information theoretic measures of neural and behavioural coupling predict representational drift

PLOS Computational Biology

Dear Dr. O'Leary,

Thank you for submitting your manuscript to PLOS Computational Biology. After careful consideration, we feel that it has merit but does not fully meet PLOS Computational Biology's publication criteria as it currently stands. Therefore, we invite you to submit a revised version of the manuscript that addresses the points raised during the review process.

Please submit your revised manuscript within 60 days Aug 12 2025 11:59PM. If you will need more time than this to complete your revisions, please reply to this message or contact the journal office at ploscompbiol@plos.org. Please include the following items when submitting your revised manuscript:

We look forward to receiving your revised manuscript.

Kind regards,

Stefano Panzeri

Academic Editor

PLOS Computational Biology

Daniele Marinazzo

Section Editor

PLOS Computational Biology

**Additional Editor Comments:**

Dear Tim, as you see the paper generated considerable interested but it also requires additional work, which seems feasible within a standard revision cycle. Please let me know if you have questions about the process. Stefano

**Journal Requirements:**

2) Your manuscript is missing the following sections: Results.  Please ensure all required sections are present and in the correct order. Make sure section heading levels are clearly indicated in the manuscript text, and limit sub-sections to 3 heading levels. An outline of the required sections can be consulted in our submission guidelines here:

Potential Copyright Issues:

i) Figure 1A. Please confirm whether you drew the images / clip-art within the figure panels by hand. If you did not draw the images, please provide (a) a link to the source of the images or icons and their license / terms of use; or (b) written permission from the copyright holder to publish the images or icons under our CC BY 4.0 license. Alternatively, you may replace the images with open source alternatives. See these open source resources you may use to replace images / clip-art:

5) This link https://github.com/kris-heiney/PID-Drift/ reaches a 404 error page. Please amend this to a new link or provide further details to locate the data.

6)  Please ensure that the funders and grant numbers match between the Financial Disclosure field and the Funding Information tab in your submission form. Note that the funders must be provided in the same order in both places as well.

7) Please send a completed 'Competing Interests' statement, including any COIs declared by your co-authors. If you have no competing interests to declare, please state "The authors have declared that no competing interests exist". Otherwise please declare all competing interests beginning with the statement "I have read the journal's policy and the authors of this manuscript have the following competing interests:"

**Reviewers' comments:**

Reviewer's Responses to Questions

**Comments to the Authors:**

Reviewer #1: The review is uploaded as an attachment

Reviewer #2: This paper analyzes two existing Ca imaging datasets using partial information decomposition methods, investigating the relationship between synergy, redundancy, and representational drift in the neural code for the areas and tasks for which the data is available. The core result of the paper is that redundancy and representational stability are strongly correlated across neurons. This result is interpreted as to suggest that drift may be organized across neural populations to preserve encoding of important task variables, and that measuring redundancy can help understanding this organization.

Overall, the idea is interesting and the evidence presented for the correlation between redundancy and stability is convincing. However, I have multiple concerns with the claims made by the paper that prevent me from recommending it for acceptance in its current state.

1 Major

═══════

The discussion contains some statements that are not backed up by the evidence presented in the Results section. This is particularly problematic because these appear to be the key conclusions that the paper attempts to draw from the results.

1.1 Existence of redundant cliques

──────────────────────────────────

> Our results demonstrate that the most stable cells in a population tend to participate in redundantly connected cliques that in turn synergistically connect to the rest of the population. Additionally, among these highly redundant subpopulations, those neurons with greater synergistic coupling to the population tend to be less stable

I do not see how this follows from what was presented above. There is nothing in the results about cliques (or other network/graph theory concepts), how they may be defined or measured, or their tendency to be synergistically connected to the rest of the population. The closest thing I can find is the plot in figure 2d, which is an example for one mouse in one recording session, and that anyway is only an illustration. There is no attempt at formalizing what it would mean for such cliques to exist and showing some quantitative evidence for that hypothesis, or that synergistic connections are indeed more common between neurons belonging to different cliques (as opposed, I assume, to between neurons within the same clique, or pairs of neurons such that one belongs to a clique but the other doesn't, or pairs of neurons that do not belong to any clique). Overall, this passage reads to me as a bit of (interesting!) speculation that should either be clearly marked as such, therefore narrowing the scope of the paper's conclusions, or developed more into an hypothesis that can actually be tested.

1.2 Possible origins of redundant coupling

──────────────────────────────────────────

> A closer look at select pairs of neurons suggests that redundant coupling may arise from either structural connections or tuning relationships among other unobserved variables.

I do not see where this closer look is taken in the Results. What selected pairs of neurons? What is the evidence presented for these pairs of neurons supporting the hypothesis that redundancy is due to structural connections or tuning relationships among other variables? and finally, these seem like very broad scenarios, so broad to be borderline uninformative: what are some alternative hypotheses? in other words, if redundancy is not due to either structural connections or tuning relationships, what else could it be due to? it would be good to provide some examples to clarify why this is an interesting hypothesis to explore rather than just a general property of redundant random variables connected in a causal network, which could either share common inputs or have direct interactions.

1.3 Unclear support for the idea of "generally salient task features"

─────────────────────────────────────────────────────────────────────

> [..] we conjecture that redundancy reveals how drift is constrained across the population in a way that limits interference with relationships among task variables that are more generally relevant across many tasks. […] this work provides compelling evidence that stable subsets of the population encode for generally salient task features, evident in the stimulus-conditioned relationships among the neuronal responses.

I have multiple issues with this passage. First, I think it is not clear what exactly is meant by "generally salient task features" here, which is not a good thing given that the passage claims that the paper offers compelling evidence for something involving those features. What is an example of a generally salient task feature? Second, my best guess for its meaning from the broader context in the paragraph is that "generally salient features" is just another way of saying "task variables that are generally relevant across many tasks", which is a much clearer concept. Assuming that is the case, I do not understand how the present work could possibly provide any evidence (let alone "compelling" evidence) for this, given that, of the two experiments analyzed, one had only one task and the other had no task at all. Either this claim is vastly overstated, or there is some key logical passage not well explained in the results, that from the distribution of the regression coefficients in figures 2b,c (essentially the only relevant pieces of quantitative evidence) allows to jump to what seems like such a far-fetched conclusion.

2 Minor

═══════

2.1 Clarity of the central claim

────────────────────────────────

There seems to be some misalignment in the emphasis placed on different claims between the abstract and the discussion. As mentioned above, the discussion places emphasis on the differences between how redundancy and synergy are distributed in the neural population, while the abstract reads as follows:

> We found that a neuron’s tuning stability is positively correlated with the strength of its average pairwise redundancy with the population, and that these high-redundancy neurons also tend to show high average pairwise synergy

Picking one central claim, sticking by it and ensuring it is well supported by the data would make the paper much more impactful in my view.

2.2 Oversimplified explanation/example of synergy

─────────────────────────────────────────────────

The following passage seems oversimplified:

> For example, a pair of neurons with shared tuning to a second external variable v would show high redundancy, as would a pair of neurons that mutually excite one another. In contrast, a pair of neurons with dissimilar tuning to v and a mutually inhibiting pair would show high synergy.

While the statements about redundancy seem correct, that on synergy seems potentially misleading. I agree with the sentiment that it is easy to construct toy examples of mutually-inhibiting model neurons that code synergistically for some external variable, but this does not immediately imply that such pairs always have high synergy. Consider the following example, where v is taken as an external variable and x, y are the activity states of two neurons:

━━━━━━━━━━━━━━━━━━━━

v x y p(x,y|v)

────────────────────

0 0 0 3/4

0 0 1 1/4

0 1 0 0

0 1 1 0

────────────────────

1 0 0 3/4

1 0 1 0

1 1 0 1/4

1 1 1 0

━━━━━━━━━━━━━━━━━━━━

In this scenario, the neurons are mutually inhibiting in the sense that the (1,1) state never occurs, and they have "dissimilar tuning" in the sense that x is tuned for v=1 and y is tuned for v=0. But it doesn't seem like there should be any synergy, because for both values of the variable v there is one of the two neurons such that observing that neuron will determine the value of v with certainty (in other words, there is no information in the system that is accessible only by simultaneous observation of the two neurons).

If this counterexample is correct, I don't think that this passage is particularly helpful for the general reader who may not be familiar with PID. I suggest simply modifying the passage to state that neurons with mutual inhibition may exhibit synergy, perhaps adding a couple of toy examples in a supplemental figure to show when this may or may not be true.

2.3 Statistical impact of multiple-counting of neurons recorded in different days

─────────────────────────────────────────────────────────────────────────────────

If I understand correctly, for the analyses in the paper (for example in the regression) a neuron recorded over multiple days is counted as multiple, statistically independent datapoints. If this is the case, this seems like a very strong assumption at best; it should be at least highlighted in the methods and its potential implications on the conclusions drawn from the regression analyses discussed.

2.4 Details on statistical inference

────────────────────────────────────

The paper mentions that the regression analysis is performed using elastic net with α=0.5, but several key details are missing.

• what is the reason for chosing this particular value of α?

• how is λ selected? the methods say "We selected the λ parameter corresponding to the model with the lowest mean squared error to obtain the final coefficient values", but this is ambiguous. This process probably involved cross-validating the MSE on some held-out data (which is the appropriate way of doing this), but if so it should be stated, and the details of the cross-validation procedure should be given.

• the elastic net paper by Zou and Hastie should be cited.

Moreover, Figure 1 caption mentions correctly that the three predictors considered all have strong pairwise correlations with each other. Unfortunately the reader is left to read this in a fairly contrived way from Figure 1a, while it could be made evident at a glance in a simple pairplot. Please provide a pairplot of the predictors (for instance as a supplementary figure).

2.5 Impact of binning:

──────────────────────

Around line 301 the text says: "Limitations of the data indicated a coarse bin size, which we verified was sufficient to support our conclusions."

It is not immediately clear from this passage how this was verified, and how it would have been possible to detect if it was otherwise. Please expand.

2.6 Typos / small edits

───────────────────────

• Figure 1 caption: "We binned the position is binned into…"

• lines 171-172: "Fig 2(b) and (c) shows the regression coefficients for the three regressors for PPC tuning to position in the maze and heading of the animal and for V1 tuning to the drifting gratings and naturalistic movies, respectively." This passage should probably read "…for PPC tuning *stability*…", etc.

• Please cite the Dryad repo also for Marks and Goard.

Reviewer #3: See attachment

**Have the authors made all data and (if applicable) computational code underlying the findings in their manuscript fully available?**

Reviewer #1: **No:** Github link is broken but I am sure they will do.

Reviewer #2: Yes

Reviewer #3: **No:** The repository with the code does not exist

PLOS authors have the option to publish the peer review history of their article (what does this mean?). If published, this will include your full peer review and any attached files.

Reviewer #1: No

Reviewer #2: No

Reviewer #3: No

**Figure resubmission:**
---

## [Decision Letter · Decision Letter 1]

11 Dec 2025

PCOMPBIOL-D-25-00928R1

Information theoretic measures of neural and behavioural coupling predict representational drift

PLOS Computational Biology

Dear Dr. O'Leary,

Thank you for submitting your manuscript to PLOS Computational Biology. After careful consideration, we feel that it has merit but does not fully meet PLOS Computational Biology's publication criteria as it currently stands. Therefore, we invite you to submit a revised version of the manuscript that addresses the points raised during the review process.

We look forward to receiving your revised manuscript.

Kind regards,

Stefano Panzeri

Academic Editor

PLOS Computational Biology

Daniele Marinazzo

Section Editor

PLOS Computational Biology

**Reviewers' comments:**

Reviewer's Responses to Questions

**Comments to the Authors:**

Reviewer #1: I think the authors made a great work in addressing my comments and concerns, and the paper largely improved.

I have two remarks left:

Lines 106–109 of the revised manuscript, in which examples of synergy and redundancy are proposed, might be confusing. A pair of neurons with dissimilar tuning might generate synergy, but the example in Fig 1 involves neurons with similar tuning (both towards 0). I think the easiest way to explain this to a naive reader is through noise correlations at a fixed stimulus (as done in Fig 1 indeed).

I would consider a third figure combining the right part of Fig 2 and Fig S4, to give more emphasis to the network result, which is quite nice in my opinion.

Overall, I congratulate the authors for the work done and I am favourable to publication in PLOS Computational Biology.

Reviewer #2: I thank the authors for their reply, the edits to the text and the additional analyses which help clarifying some of my concerns. I still have some issues with the regression and clique analyses, which I detail below.

1. Regression analyses

A. what is the motivation for performing a stepwise regression? why not simply add the interaction coefficients to the elastic net models, as suggested by R1?

B. The text says "In the PPC dataset with U the position, when the redundancy index × synergy index term was present, it tended to be negative". I do not see how one can conclude that from Figure S2 (for instance looking at the pooled data for all mice). Figure S2 shows that the red:syn interaction is generally zero, and that the MI:red interaction is generally negative. Looking at the other plots in figure S3 it is hard to convince oneself that there is anything that makes the distribution of the red:syn parameter estimates of PPC/position more interesting than the other interaction terms. Can you clarify this point?

C. Even disregarding the previous point, I don't fully agree with the interpretation of the results of the regression with interaction terms. Consider the sentence "among neurons with a given redundancy index, a neuron with a higher synergy index would tend to exhibit lower tuning stability". The correct interpretation of the interaction term is that, if the interaction is negative, it means that if you partition the neurons in low-redundancy and high-redundancy then the high-redundancy ones will have a synergy-stability relationship with a smaller slope than the ones with low redundancy. But this is not the same thing as saying that the slope must be negative ("a neuron with a higher synergy index would tend to exhibit lower tuning stability"). Let me expand. Consider, in general, a linear model with two predictors, X1 and X2, and target variable Y. Then the model with an interaction is Y=β₁X₁+β₂X₂+β₃X₁X₂, so that when one fixes X₂ (the redundancy in our case) one is left with Y=(β₁+β₃X₂)X₁+β₂X₂. This is a linear relationship with slope (β₁+β₃X₂) and intercept β₂X₂. So whether the slope is positive or negative (that is, whether neurons with higher synergy will have lower stability) depends on both β₁ and β₃. In the case at hand, β₁ is generally negative, but it would be better to specify in the text that the effect holds because of the interplay of the two terms, especially as this is important to keep in mind when trying to interpret the other plots in figure S3.

D. Something I missed in the first round is related to the following sentence in the methods: "If any predictor did not have a significant correlation with the stability in a single-variable model (i.e., p > 0.05), its regression coefficient was set to 0." Can you expand on this? My understanding is that (a) Elastic Net does not allow for a closed-form sampling distribution (even an asymptotic one) of the parameter estimates under the null hypothesis, so I don't understand how these p-values were computed, unless there was something more complex going on that was not described in the Methods (such a permutation test of some sort). (b) it is highly unusual to manually threshold parameter estimates in this way, at least in elastic net, because the L1 component of the regularizer should automatically take care of setting irrelevant parameters exactly to zero. So I struggle to see the statistical rationale behind this operation.

2. Clique analysis

A. If I understand correctly, the p-values reported in Table S1 come from computing a Wilcoxon signed-rank test to the data that is displayed as dots in Figure S4 b,d. Each datapoint is two-dimensional and it is calculated as (for instance) the stability of the nodes in redundant and synergistic cliques that can be found in a session. For each session there are many datapoints because each of them corresponds to a different choice of the binarization threshold for the graph, used in finding the cliques. One problem I see with this procedure is that the datapoints thus determined are clearly not statistically independent, and thus do not satisfy the assumptions of the Wilcoxon test. To show why this is the case, imagine what happens if the increments of the threshold "grid" were to become very small: at some point one would start getting datapoints that are identical copies of each other, because in general the binarization-and-clique-finding process will have some degree of robustness to the choice of binarization process (meaning that the cliques in the binarized network, and essentially the network itself, will be the same for threshold t and t+ε, with ε small). Whether there are choices of the increments for this parameter that guarantee that networks obtained with different thresholds can be considered as independent random samples from some underlying distribution is not obvious to me, a priori.

B. Tangentially, table S4 contains 32 statistical tests and I couldn't find a mention of the multiple-test correction that was used.

C. I get the reasoning behind studying the connectivity between nodes in a clique and the nearest neighbors of the clique, but I don't understand why this is studied always in the opposing graph (synergistic graph for the redundant cliques and redundant graph for the synergistic cliques). This should be explained in the text.

D. Regarding the summary sentence for the clique results paragraph: "This indicates that highly stable neurons tend to redundantly couple to each other and synergistically couple to other lower-stability neurons in the population". I think the results, at face value (modulo my concerns above), show that neurons in redundant cliques tend to have higher stability. I don't think this is the same as saying that stable neurons tend to redundantly couple with each other. Is it possible, for instance, that highly stable neurons also tend to participate in very few cliques? in that scenario, it would not be correct to say that highly stable neurons tend to connect to each other in such way, because they primarily would simply tend to connect less (or connect in ways that do not form cliques). Perhaps the data in the other figures already excludes this possibility, but if it is the case it would be good to say that explicitly. More generally, the sentence I am commenting on here tries to draw a conclusion on how stability influences redundancy and synergy, while the analyses go in the opposite logical direction, so it seems that some additional care is required.

3. Possible origin of redundant coupling

Thank you, I think the additional text and supplementary figures help conveying a clearer message.

4. Unclear support for the idea of "generally salient task features"

Thank you for changing the language in this passage to better highlight that this is a conjecture, rather than a conclusion of the work.

5. Minor points from my original review

Thank you for making these changes, and in particular for the addition of the toy example in the supplement, which I find quite informative. Clearly I had misinterpreted that passage on my first read, but hopefully this additional example will help other readers too.

Reviewer #3: The reviesed version of the manuscript addressed the concerns I raised. The manuscript has significantly ameliorated and I suggest considering publication.

Best.

**Have the authors made all data and (if applicable) computational code underlying the findings in their manuscript fully available?**

Reviewer #1: Yes

Reviewer #2: Yes

Reviewer #3: Yes

PLOS authors have the option to publish the peer review history of their article (what does this mean?). If published, this will include your full peer review and any attached files.

Reviewer #1: **Yes:** Simone Blanco Malerba

Reviewer #2: No

Reviewer #3: No

**Figure resubmission:**
---

## [Editor Report · Decision Letter 2]

4 Feb 2026

Dear DR O'Leary,

We are pleased to inform you that your manuscript 'Information theoretic measures of neural and behavioural coupling predict representational drift' has been provisionally accepted for publication in PLOS Computational Biology.

Best regards,

Stefano Panzeri

Academic Editor

PLOS Computational Biology

Daniele Marinazzo

Section Editor

PLOS Computational Biology

---

## [Editor Report · Acceptance letter]

PCOMPBIOL-D-25-00928R2

Information theoretic measures of neural and behavioural coupling predict representational drift

Dear Dr O'Leary,

I am pleased to inform you that your manuscript has been formally accepted for publication in PLOS Computational Biology. Your manuscript is now with our production department and you will be notified of the publication date in due course.

With kind regards,

Anita Estes
